# Identification of immune-related mechanisms of cetuximab induced skin toxicity in colorectal cancer patients

Jin Hyun Park[1], Mi Young Kim[1], In Sil Choi[1], Ji-Won Kim[2], Jin Won Kim[2], Keun-Wook Lee[2], Jin-Soo Kim[1]*

1 Department of Internal Medicine, Seoul Metropolitan Government Seoul National University Boramae Medical Center, Seoul National University College of Medicine, Seoul, South Korea, 2 Department of Internal Medicine, Seoul National University Bundang Hospital, Seoul National University College of Medicine, Seongnam, South Korea

* gistmd@gmail.com

**Data Availability Statement:** All relevant data are within the paper and its Supporting Information files.

## Abstract

Skin rash is a well-known predictive marker of the response to cetuximab (Cmab) in metastatic colorectal cancer (mCRC). However, the mechanism of skin rash development is not well understood. Following exposure to EGFR-targeted therapies, changes in IL-8 levels have been reported. The aim of this study was to evaluate the association between skin rash and inflammatory cytokine levels, including IL-8. Between 2014 and 2017, we prospectively enrolled 38 mCRC patients who underwent chemotherapy with either Cmab or bevacizumab (Bmab) at two hospitals. We performed multiplex cytokine ELISA with 20 inflammatory cytokines including E-selectin, GM-CSF, IFN-alpha, IFN-γ, IL-1 alpha, IL-1 beta, IL-4, IL-6, IL-8, IL-10, IL-12p70, IL-13, IL-17A, IP-10, MCP-1, MIP-1 alpha, MIP-1 beta, P-selectin, sICAM-1, and TNF-alpha at baseline before cycle 1, 24 h after cycle 1, before cycle 2 (= 14 d), and before cycle 3 (= 28 d). Cytokine levels were compared using ANOVA after log-transformation. IL-8 genotypes in 30 patients treated with Cmab were determined using the polymerase chain reaction restriction fragment length polymorphism technique. Depending on the RAS mutational status, 30 and eight patients were treated with Cmab and Bmab-based chemotherapy, respectively. Skin rash developed in 23 (76.6%) of the 30 patients treated with Cmab plus FOLFIRI, after cycle 1. Only the mean log-transformed serum IL-8 level in patients with skin toxicity was statistically lower (2.83 ± 0.15) than in patients who did not experience skin toxicity (3.65 ± 0.27) and received Bmab (3.10 ± 0.26) (ANOVA test, p value = 0.0341). In addition, IL-8 polymorphism did not affect IL-8 levels, skin toxicity, or tumor response in Cmab treated patients. This study suggests that the inflammatory cytokine levels might be affected by Cmab exposure and are associated with the development of skin rash in mCRC patients. Further studies are warranted to evaluate this interaction in Cmab treated patients.

**Funding:** This study was supported by grant no. 0320150400 from the Seoul National University Hospital Research Fund. The funders had no role in study design, data collection and analysis, decision to publish, or preparation of the manuscript.

**Competing interests:** The authors have declared that no competing interests exist.

## Introduction

Colorectal cancer is the third most commonly diagnosed cancer worldwide, and 551,269 deaths per year were reported globally in 2018 [1]. Several agents are now available for the treatment of newly diagnosed metastatic colorectal cancer (mCRC), including 5-fluorouracil (5-FU)-based chemotherapy with targeted agents. Folinic acid, infusional 5-FU and irinotecan (FOLFIRI) is one of the most commonly used first-line and second-line chemotherapy for mCRC, and is usually administered with vascular endothelial growth factor (VEGF)- or epidermal growth factor receptor (EGFR)-targeted agents. Cetuximab (Cmab) is a chimeric human/mouse IgG1 monoclonal antibody that targets EGFR, a cell surface receptor overexpressed in several types of cancer. Median overall survival (OS) for RAS wild-type (wt) mCRC patients treated with FOLFIRI plus Cmab exceeded 20 months [2–4].

Although the integration of these novel agents considerably improved the outcome of mCRC patients, the use of targeted agents leads to greater treatment-related toxicity. The major adverse effects associated with Cmab are skin toxicity, including acneiform rash, xerosis, pruritus, and nail changes [5, 6]. Skin toxicity can adversely affect patients' quality of life and treatment compliance. Treating physicians would reduce the dose and even discontinue chemotherapy because of these toxicities [7], thus possibly affecting the treatment outcome of anti-EGFR therapy. These skin toxicities have been identified as class-specific adverse events associated with anti-EGFR agents.

Notably, anti-EGFR agent-induced rash can be a useful surrogate predictive marker for a substantially improved OS, progression-free survival (PFS), and tumor response to several EGFR inhibitors approved for clinical use [8]. Although the pathophysiology of Cmab-induced skin toxicity remains elusive, multiple EGFR-dependent homeostatic functions of the skin can lead to skin toxicity from Cmab [6]. A previous study investigated the possible mechanisms of EGFR inhibition-associated cytokine production in keratinocytes and patients treated with EGFR inhibitors [9]. Proteomic analysis of mCRC patients treated with Cmab showed downregulation of phospho(p)-EGFR, p-MAPK, and proliferation and substantial upregulation of p27$^{Kip1}$ and p-STAT3 levels [10].

IL-8 is a member of the cytokine family and plays a key role in neutrophil recruitment and degranulation [11]. IL-8 primarily mediates the activation and migration of neutrophils from peripheral blood into tissues and is involved in the initiation and amplification of inflammatory responses by the immune system. Following the incubation of human epidermal keratinocytes with an EGFR inhibitor, decreased IL-8 levels were observed in keratinocytes [9]. In another study, skin rash induced by EGFR inhibitors was ameliorated by neutralization of IL-8 with a specific mAb [12]. Moreover, IL-8 polymorphisms have been associated with the risk of developing several types of cancer in cohort studies [13]. The IL-8 gene is located on chromosome 4q13-21 in humans, and is composed of four exons, three introns, and a promoter region. Three common polymorphisms have been studied in the IL-8 gene: -251 A/T, +396 G/T, and +781 C/T [14, 15]. Several studies have shown that IL-8 polymorphisms are associated with the risk of developing lung, stomach, breast, and ovarian cancers [13, 16–18]. In colon cancer, IL-8 has been reported to play a role in promoting colon cancer growth, progression, and metastasis [19, 20]. Furthermore, several studies have suggested that the IL-8-251T (AT + TT) allele is associated with increased IL-8 production [23, 24].

IL-8 polymorphisms may serve as predictive biomarkers for Bmab-based chemotherapy outcomes in RAS mutant mCRC patients [21]. In this study, carriers of the IL-8 allele c.-251TA+AA showed shorter PFS and OS compared to the TT allele. Notably, a previous study suggested that IL-8 polymorphisms influence IL-8 levels [22]. There have been a few reports that the IL-8 genotype could be associated with different IL-8 production in patients [23, 24].

We aimed to investigate the association between the dynamic changes in inflammatory cytokines, including IL-8, and the development of skin rash after Cmab exposure in colorectal cancer patients. We also wanted to evaluate whether IL-8 polymorphisms could affect the level of IL-8, skin rash development, and tumor response in mCRC patients treated with Cmab.

## Materials and methods

### Patients

Between 2014 and 2017, we prospectively enrolled 38 patients with mCRC who were treated with FOLFIRI with either Cetuximab (Erbitux; Merck-Serono, Darmstadt, Germany) or bevacizumab (Bmab) at two academic hospitals. This study was approved by the institutional review boards of SMG-SNU Boramae Medical Center (IRB No.26-2015-39) and Seoul National University Bundang Hospital (IRB No. B-1507/306-404). The study protocol conformed to the ethical guidelines of the 1975 Declaration of Helsinki, as reflected in a priori approval by the institution's human research committee. Written informed consent was obtained prior to study procedures. We obtained tumor specimens and performed pyrosequencing to detect KRAS (codons 12, 13, and 61) and NRAS mutations (codons 12,13, and 61). Of 38 patients, seven with KRAS or NRAS-mutated tumors and one whose tumor did not express EGFR by immunohistochemistry were treated with FOLFIRI plus Bmab. Thirty patients with wild-type KRAS/NRAS tumors received FOLFIRI with Cmab.

### Treatment and samples

In the Cmab and Bmab groups, patients received a 500 mg/m$^2$ Cmab infusion over 2 h and 5 mg/kg Bmab infusion over 90 min on day 1, respectively. The FOLFIRI regimen consisted of 180 mg/m$^2$ irinotecan in a 2 h intravenous infusion on day 1, followed by 400 mg/m$^2$ leucovorin over 2 h, before 400 mg/m$^2$ 5-FU as an intravenous bolus injection, and 2400 mg/m$^2$ 5-FU as a 46 h infusion immediately after the 5-FU bolus injection on day 1. We obtained blood samples and measured cytokine levels at baseline before cycle 1, 24 h after cycle 1, before cycle 2 (= 14 d), and before cycle 3 (= 28 d). Genomic DNA was extracted from 2 mL blood samples from all patients using the DNeasy Blood & Tissue Kit (QIAGEN,Hilden, Germany).

### Genotyping analysis

IL-8 genotyping was performed using a polymerase chain reaction–restriction fragment length polymorphism analysis (PCR–RFLP). Reaction conditions for the PCR were 95°C for 5 min for the initial denaturation, followed by 35 cycles of denaturation at 95°C for 30 s, annealing with the primer specific temperature (Table 1) for 60 s, extension at 72°C for 60 s, a final extension for 10 min at 72°C, followed by final cooling at 4°C. The primers used are listed in Table 1. PCR was performed in a total reaction volume of 25 μL containing 100 ng DNA, 0.2 mM of each dNTP, 2.0 mM MgCl2, and 1.25 U Taq DNA polymerase (Takara Bio, Shiga, Japan). Following purification with the QIAquick PCR Purification Kit (QIAGEN), PCR products were digested with 5 units of restriction enzymes for 3 h at 37°C. The restriction enzymes used are listed in Table 1. Digested PCR products were separated by gel electrophoresis using a 2%–3% agarose gel. Bands were visualized using ethidium bromide staining. A gel electrophoresis pattern of polymorphism in each locus is presented in Fig 1, showing the resulting patterns after restriction enzyme digestion for all three possible genotypes (two homozygous and one heterozygous).

**Table 1. Primers for amplification (primer sequence (5'→ 3'), annealing temperature and restriction enzyme for digestion).**

| No. | Polymorphism | Primer sequence (5' → 3') | Annealing temperature | Restriction enzyme | Product size (bp) |
|---|---|---|---|---|---|
| 1 | IL8-251 T/A | F: TCATCCATGATCTTGTTCTAA (21) | 55 | MfeI | T: 524 |
| | | R: GGAAAACGCTGTAGGTCAGA (20) | | | A: 449, 75 |
| 2 | IL8+781 C/T | F: CTCTAACTCTTTATATAGGAATT (23) | 50 | EcoRI | T: 203 |
| | | R: GATTGATTTTATCAACAGGCA (21) | | | C: 184, 19 |
| 3 | IL8+1633 C/T | F: CTGATGGAAGAGAGCTCTGT (20) | 55 | NIaIII | T: 397 |
| | | R: TGTTAGAAATGCTCTATATTCTC (23) | | | C: 234, 163 |
| 4 | IL8+2767 A/T | F: CCAGTTAAATTTTCATTTCAGGTA (24) | 50 | BstZ17I | A: 222 |
| | | R: CAACCAGCAAGAAATTACTAA (21) | | | T: 198, 24 |

## Cytokine measurements

The Human Inflammation Panel kit (EPX200-12185-901, Affymetrix eBioscience, Vienna, Austria) was used to analyze 20 inflammatory mediators (including E-selectin, GM-CSF, IFN-alpha, IFN-gamma, IL-1 alpha, IL-1 beta, IL-4, IL-6, IL-8, IL-10, IL-12p70, IL-13, IL-17A, IP-10, MCP-1, MIP-1 alpha, MIP-1 beta, P-selectin, sICAM-1, and TNF-α) in plasma samples. Plasma samples were thawed at 4°C, diluted 1:2 with Universal Assay Buffer (EPX-11111-000, Affymetrix eBioscience, Vienna, Austria) and assayed according to the manufacturer's instructions. Following performing measurements with a Bio-Plex 200 (Luminex 200) system, samples were analyzed using Bio-Plex Manager (4.1.1) (both from Bio-Rad Laboratories Inc, Hercules, CA).

## Statistical analysis

R (version 4.0.3; R Foundation for Statistical Computing, Vienna, Austria) and SPSS version 26 (IBM, Armonk, NY) for Windows statistical software were used for statistical analyses. For normality assumption for cytokines, the Shapiro-Wilk test was applied. Differences in cytokine levels between groups were analyzed using analysis of variance (ANOVA) or the Kruskal-Wallis test. The Hardy-Weinberg equilibrium assumption was assessed for all IL-8 polymorphisms. Based on the IL-8 polymorphism, the level of IL-8 was analyzed using the Kruskal-Wallis test and Mann-Whitney U test. According to IL-8 genotyping, skin toxicity and tumor

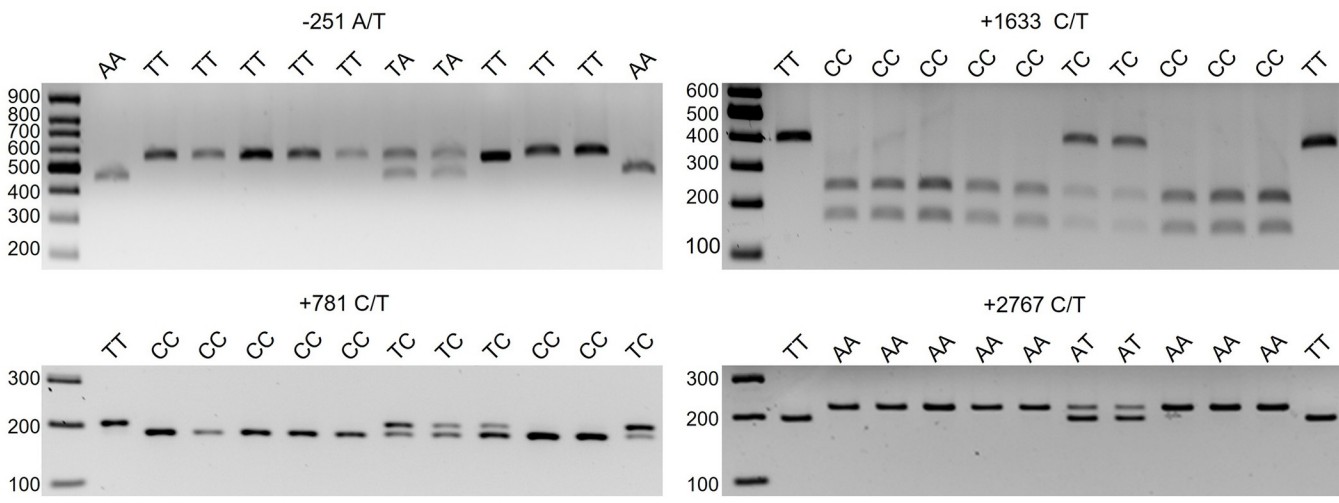

**Fig 1. Polymerase chain reaction–restriction fragment length polymorphism (PCR-RFLP) agarose gel electrophoresis patterns of IL-8 polymorphisms.**

response were analyzed using the Chi-square test, Fisher's exact test, and Cochran-Armitage trend test. All p-values were two-tailed. Statistical significance was set at $P < 0.05$.

## Results

### Baseline characteristics

Between 2014 and 2017, 47 patients with mCRC were prospectively enrolled. Baseline serum samples were obtained from these patients before chemotherapy, and IL-8 genotyping was analyzed. Following the first cycle of chemotherapy, we performed serial analysis of 20 inflammatory mediators with Human Inflammation Panel kit in only 38 patients, because nine of the 47 patients refused subsequent blood sampling owing to worsening of the disease or patient discretion. The baseline characteristics of the 38 patients (28 men and 10 women) are presented in Table 2. The median age of the patients was 59 years (range, 37–78 years).

The median (OS) was 1064 days (95% confidence interval, 438~1689 days) for all patients. For the patients treated with Cmab, there was no statistically significant difference in terms of OS between patients with skin rash and whom without skin rash (1064 vs 888 days) (p = 0.840). The median OS in patients with skin rash grade 3, 2 and 1 were 1725, 1275 and 807 days, respectively (p = 0.528). Even though the presence or grades of skin rash were

**Table 2. Baseline characteristics of 38 patients.**

| Variable | Number | Percentage (%) |
|---|---|---|
| Age at diagnosis | | |
| Median (range) | 59 (37–78) | |
| Sex | | |
| Male | 28 | 73.7 |
| Female | 10 | 26.3 |
| Location | | |
| Left colon | 31 | 81.6 |
| Right colon | 7 | 18.4 |
| Histology | | |
| Well differentiated | 3 | 8.1 |
| Moderately differentiated | 31 | 83.8 |
| Poorly differentiated | 2 | 5.4 |
| Mucinous | 1 | 2.7 |
| Unknown | 1 | |
| KRAS mutation status | | |
| Mutation | 7 | 18.4 |
| Wild type | 31 | 81.6 |
| NRAS mutation status | | |
| Mutation | 0 | 0 |
| Wild type | 31 | 100 |
| Unknown | 7 | |
| BRAF mutation status | | |
| Mutation | 3 | 23.1 |
| Wild type | 10 | 76.9 |
| Unknown | 13 | |
| Targeted agent | | |
| Cetuximab | 30 | 78.9 |
| Bevacizumab | 8 | 21.1 |

associated with numerically improved median OS as previous reported in many clinical studies, these findings were not statistically significant due to small number of the patients.

## Cytokine levels

We compared the mean log-transformed cytokine levels among the three groups at baseline before cycle 1, 24 h after cycle 1, before cycle 2 (= 14 d), and before cycle 3 (= 28 d): group 1 (patients with Cmab-induced skin toxicity), group 2 (patients without Cmab-induced skin toxicity), and group 3 (patients treated with Bmab). From the results, we identified significant differences in IL-8 and IL-1 levels between groups 1 and 2. At baseline, the mean log-transformed serum IL-8 level in group 1 was statistically lower than that in the other two groups (2.83 ± 0.15, 3.65 ± 0.27, and 3.10 ± 0.26, p = 0.0341). We showed the mean log-transformed serum IL-8 levels at three time points for each group (Fig 2). The mean serum IL-1 beta level for patients in group 1 was lower than that for patients in the other two groups (1.94 ± 0.21, 2.98 ± 0.38, and 2.12 ± 0.36, p value = 0.0383) at 24 h after cycle 1. Also, the mean serum IL-1 alpha and MIP-1 beta levels for patients in group 3 was lower than that for patients in the other two groups at 24 h after cycle 1, but the levels were not different between patient group 1 and 2. Detailed results are presented in Table 3.

## The effect of the IL-8 polymorphism

We compared tumor responses and IL-8 genotypes in 30 patients treated with Cmab. Allele -251 is known to have complete linkage disequilibrium with +1633 and +2767. The mean log-transformed serum IL-8 levels by IL-8 genotypes are presented in Table 4 and Fig 3. Serum IL-8 levels were not significantly different between IL-8 genotypes in this cohort. For

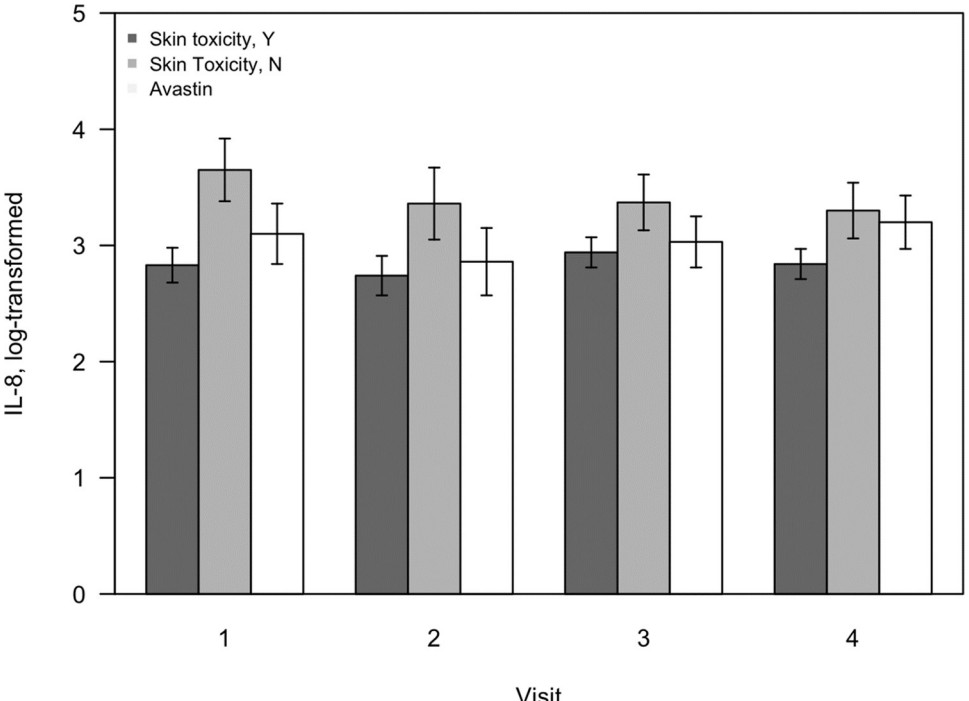

**Fig 2. The mean log-transformed serum IL-8 level at three time points for each group.**

**Table 3. Log-transformed cytokine levels of the three groups at different time points.**

| | Time points | Group (Least squares mean ± std. error) | | | P-value |
|---|---|---|---|---|---|
| | | **1** | **2** | **3** | |
| IL-8 | 1 | 2.83 ± 0.15 | 3.65 ± 0.27 | 3.10 ± 0.26 | 0.0341 |
| | 2 | 2.74 ± 0.17 | 3.36 ± 0.31 | 2.86 ± 0.29 | 0.1248 |
| | 3 | 2.94 ± 0.13 | 3.37 ± 0.24 | 3.03 ± 0.22 | 0.3626 |
| | 4 | 2.84 ± 0.13 | 3.30 ± 0.24 | 3.20 ± 0.23 | 0.2318 |
| E-Selectin | 1 | 9.99 ± 0.11 | 9.83 ± 0.19 | 9.91 ± 0.18 | 0.7626 |
| | 2 | 10.03 ± 0.10 | 9.87 ± 0.18 | 9.80 ± 0.17 | 0.4760 |
| | 3 | 10.06 ± 0.10 | 9.79 ± 0.18 | 9.88 ± 0.17 | 0.3872 |
| | 4 | 10.04 ± 0.11 | 9.81 ± 0.19 | 9.97 ± 0.18 | 0.5593 |
| GM-CSF | 1 | 3.88 ± 0.21 | 4.25 ± 0.37 | 3.70 ± 0.36 | 0.5178 |
| | 2 | 3.95 ± 0.23 | 4.25 ± 0.41 | 3.65 ± 0.40 | 0.6175 |
| | 3 | 3.98 ± 0.16 | 4.14 ± 0.30 | 3.77 ± 0.28 | 0.7556 |
| | 4 | 4.14 ± 0.21 | 4.21 ± 0.37 | 3.87 ± 0.36 | 0.7874 |
| IFN-alpha | 1 | 1.36 ± 0.14 | 1.58 ± 0.26 | 1.42 ± 0.25 | 0.7610 |
| | 2 | 1.21 ± 0.18 | 1.48 ± 0.32 | 0.99 ± 0.30 | 0.3866 |
| | 3 | 1.49 ± 0.12 | 1.64 ± 0.22 | 1.39 ± 0.21 | 0.7675 |
| | 4 | 1.46 ± 0.13 | 1.64 ± 0.24 | 1.55 ± 0.22 | 0.8434 |
| IFN-gamma | 1 | 4.09 ± 0.29 | 4.61 ± 0.53 | 4.48 ± 0.50 | 0.6150 |
| | 2 | 4.10 ± 0.32 | 4.62 ± 0.58 | 4.16 ± 0.54 | 0.6874 |
| | 3 | 4.20 ± 0.27 | 4.64 ± 0.49 | 4.36 ± 0.45 | 0.7543 |
| | 4 | 4.26 ± 0.29 | 4.58 ± 0.53 | 4.44 ± 0.50 | 0.8531 |
| IL-1 alpha | 1 | 2.35 ± 0.12 | 2.72 ± 0.22 | 2.26 ± 0.21 | 0.3420 |
| | 2 | 2.24 ± 0.19 | 2.44 ± 0.35 | 1.49 ± 0.33 | 0.0079 |
| | 3 | 2.61 ± 0.09 | 2.90 ± 0.17 | 2.81 ± 0.16 | 0.5083 |
| | 4 | 2.57 ± 0.11 | 2.92 ± 0.20 | 2.74 ± 0.19 | 0.5514 |
| IL-1 beta | 1 | 2.00 ± 0.21 | 2.38 ± 0.38 | 2.22 ± 0.36 | 0.6014 |
| | 2 | 1.94 ± 0.21 | 2.98 ± 0.38 | 2.12 ± 0.36 | 0.0383 |
| | 3 | 2.22 ± 0.18 | 2.35 ± 0.32 | 1.90 ± 0.30 | 0.6084 |
| | 4 | 2.41 ± 0.17 | 2.37 ± 0.31 | 2.68 ± 0.30 | 0.7132 |
| IL-10 | 1 | 1.79 ± 0.20 | 2.32 ± 0.37 | 1.67 ± 0.35 | 0.3743 |
| | 2 | 1.87 ± 0.21 | 2.40 ± 0.39 | 1.66 ± 0.36 | 0.3039 |
| | 3 | 1.95 ± 0.20 | 2.19 ± 0.36 | 1.68 ± 0.34 | 0.5943 |
| | 4 | 2.05 ± 0.20 | 2.25 ± 0.37 | 2.06 ± 0.35 | 0.9255 |
| IL-12p70 | 1 | 3.67 ± 0.15 | 3.85 ± 0.28 | 3.79 ± 0.26 | 0.8121 |
| | 2 | 3.71 ± 0.17 | 3.79 ± 0.30 | 3.53 ± 0.28 | 0.7437 |
| | 3 | 3.76 ± 0.14 | 3.88 ± 0.25 | 3.75 ± 0.24 | 0.9209 |
| | 4 | 3.74 ± 0.14 | 3.92 ± 0.25 | 3.84 ± 0.23 | 0.8506 |
| IL-13 | 1 | 2.06 ± 0.14 | 2.34 ± 0.26 | 2.08 ± 0.25 | 0.6314 |
| | 2 | 1.98 ± 0.18 | 2.17 ± 0.33 | 1.68 ± 0.31 | 0.3577 |
| | 3 | 2.15 ± 0.12 | 2.34 ± 0.22 | 2.12 ± 0.20 | 0.7717 |
| | 4 | 2.16 ± 0.13 | 2.40 ± 0.23 | 2.24 ± 0.22 | 0.7357 |
| IL-17A | 1 | 3.02 ± 0.17 | 3.36 ± 0.30 | 3.27 ± 0.28 | 0.5216 |
| | 2 | 3.01 ± 0.17 | 3.43 ± 0.31 | 3.15 ± 0.29 | 0.4729 |
| | 3 | 3.10 ± 0.17 | 3.43 ± 0.30 | 3.31 ± 0.28 | 0.5790 |
| | 4 | 3.12 ± 0.16 | 3.55 ± 0.30 | 3.62 ± 0.28 | 0.2281 |

(*Continued*)

**Table 3.** (Continued)

|  | Time points | Group (Least squares mean ± std. error) | | | P-value |
|---|---|---|---|---|---|
|  |  | **1** | **2** | **3** |  |
| IL-4 | 1 | 4.28 ± 0.21 | 4.71 ± 0.39 | 4.43 ± 0.36 | 0.6041 |
|  | 2 | 4.25 ± 0.25 | 4.60 ± 0.44 | 4.07 ± 0.42 | 0.5791 |
|  | 3 | 4.36 ± 0.18 | 4.67 ± 0.34 | 4.20 ± 0.31 | 0.6583 |
|  | 4 | 4.42 ± 0.21 | 4.84 ± 0.37 | 4.55 ± 0.35 | 0.6600 |
| IL-6 | 1 | 3.96 ± 0.19 | 4.21 ± 0.35 | 3.84 ± 0.32 | 0.7240 |
|  | 2 | 3.98 ± 0.21 | 4.48 ± 0.38 | 3.84 ± 0.36 | 0.3495 |
|  | 3 | 4.05 ± 0.16 | 4.20 ± 0.30 | 3.81 ± 0.28 | 0.6986 |
|  | 4 | 4.03 ± 0.20 | 4.09 ± 0.36 | 4.02 ± 0.34 | 0.9782 |
| IP-10 | 1 | 1.79 ± 0.20 | 2.32 ± 0.37 | 1.67 ± 0.35 | 0.3743 |
|  | 2 | 1.87 ± 0.21 | 2.40 ± 0.39 | 1.66 ± 0.36 | 0.3039 |
|  | 3 | 1.95 ± 0.20 | 2.19 ± 0.36 | 1.68 ± 0.34 | 0.5943 |
|  | 4 | 2.05 ± 0.20 | 2.25 ± 0.37 | 2.06 ± 0.35 | 0.9255 |
| MCP-1 | 1 | 3.76 ± 0.18 | 4.58 ± 0.33 | 3.95 ± 0.31 | 0.1597 |
|  | 2 | 3.88 ± 0.27 | 4.23 ± 0.49 | 3.54 ± 0.45 | 0.3907 |
|  | 3 | 4.10 ± 0.16 | 4.82 ± 0.30 | 4.11 ± 0.29 | 0.2118 |
|  | 4 | 4.11 ± 0.20 | 4.58 ± 0.37 | 3.85 ± 0.36 | 0.5137 |
| MIP-1 alpha | 1 | 2.03 ± 0.11 | 2.27 ± 0.20 | 2.22 ± 0.18 | 0.4811 |
|  | 2 | 1.90 ± 0.16 | 1.87 ± 0.30 | 1.77 ± 0.28 | 0.8479 |
|  | 3 | 2.12 ± 0.08 | 2.29 ± 0.15 | 2.30 ± 0.14 | 0.6515 |
|  | 4 | 2.18 ± 0.09 | 2.33 ± 0.17 | 2.34 ± 0.17 | 0.6989 |
| MIP-1 beta | 1 | 4.77 ± 0.12 | 5.06 ± 0.21 | 4.65 ± 0.20 | 0.3943 |
|  | 2 | 4.72 ± 0.17 | 4.93 ± 0.31 | 4.16 ± 0.29 | 0.0294 |
|  | 3 | 4.94 ± 0.11 | 5.23 ± 0.21 | 4.67 ± 0.19 | 0.1944 |
|  | 4 | 4.96 ± 0.09 | 5.14 ± 0.16 | 4.93 ± 0.15 | 0.8964 |
| P-selectin | 1 | 9.86 ± 0.13 | 9.82 ± 0.23 | 9.91 ± 0.22 | 0.9589 |
|  | 2 | 9.88 ± 0.11 | 9.96 ± 0.21 | 9.89 ± 0.19 | 0.9562 |
|  | 3 | 9.95 ± 0.12 | 9.77 ± 0.23 | 9.76 ± 0.21 | 0.6307 |
|  | 4 | 10.04 ± 0.11 | 9.93 ± 0.20 | 9.93 ± 0.20 | 0.8609 |
| sICAM-1 | 1 | 12.07 ± 0.19 | 12.06 ± 0.34 | 12.39 ± 0.32 | 0.6326 |
|  | 2 | 11.97 ± 0.18 | 12.09 ± 0.32 | 12.33 ± 0.30 | 0.5898 |
|  | 3 | 12.04 ± 0.18 | 12.07 ± 0.32 | 12.43 ± 0.30 | 0.5213 |
|  | 4 | 11.95 ± 0.17 | 12.12 ± 0.31 | 12.41 ± 0.29 | 0.4804 |
| TNF-alpha | 1 | 3.95 ± 0.12 | 4.21 ± 0.22 | 4.05 ± 0.21 | 0.6035 |
|  | 2 | 3.91 ± 0.14 | 4.17 ± 0.25 | 3.80 ± 0.23 | 0.4794 |
|  | 3 | 4.07 ± 0.14 | 4.22 ± 0.25 | 4.19 ± 0.23 | 0.7954 |
|  | 4 | 4.13 ± 0.11 | 4.31 ± 0.21 | 4.19 ± 0.20 | 0.8592 |

Time points 1, before cycle 1; 2, 24 h; 3, 14 d; 4, 28 d

Group 1, patients with Cmab-induced skin toxicity; 2, patients without Cmab-induced skin toxicity; 3, patients treated with Bmab

**Table 4. Serum IL-8 level according to genotyping of IL-8.**

| SNP | Minor homozygote | Heterozygote | Major homozygote | Kruskal-Wallis test P-value |
|---|---|---|---|---|
| 251 | 23.68 [15.61, 29.01] | 22.26 [14.34, 37.65] | 18.88 [13.34, 29.58] | 0.8027 |
| 781 | 24.43 [21.57, 27.28] | 19.32 [9.03, 31.78] | 19.82 [15.15, 30.21] | 0.8259 |

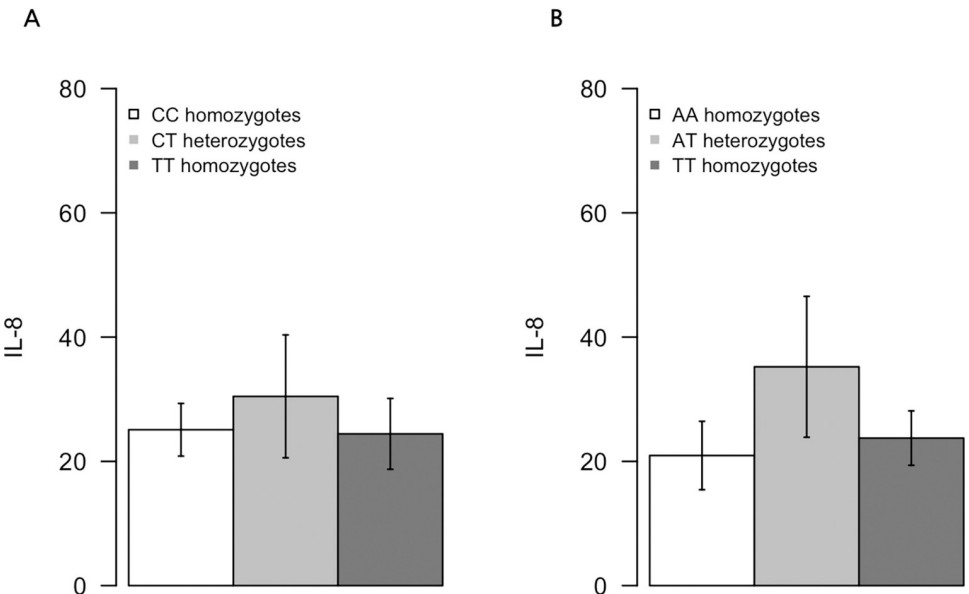

**Fig 3. Serum IL-8 level according to genotyping of IL-8.** (A) allele -251 (B) allele -781.

polymorphisms at alleles -251 and +781, tumor responses were not statistically significantly different, as presented in Table 5. Furthermore, there were no differences in skin toxicity development and severity between patients with different genotypes at alleles -251 and +781 (Table 6).

**Table 5. Tumor responses according to genotyping of IL-8.**

| SNP | Mode | | CR, PR | SD | P-value |
|---|---|---|---|---|---|
| 251 | Codominant | AA | 4 (12.90%) | 0 (0.00%) | 0.3891 |
| | | TA | 13 (41.94%) | 5 (38.46%) | |
| | | TT | 14 (45.16%) | 8 (61.54%) | |
| | Additive | AA (2) | 4 (12.90%) | 0 (0.00%) | 0.1820 |
| | | TA (1) | 13 (41.94%) | 5 (38.46%) | |
| | | TT (0) | 14 (45.16%) | 8 (61.54%) | |
| | Dominant | AA or TA (1) | 17 (54.84%) | 5 (38.46%) | 0.3216 |
| | | TT (0) | 14 (45.16%) | 8 (61.54%) | |
| | Recessive | AA (1) | 4 (12.90%) | 0 (0.00%) | 0.3024 |
| | | TA or TT (0) | 27 (87.10%) | 13 (100%) | |
| 781 | Codominant | TT | 2 (6.45%) | 0 (0.00%) | 0.4612 |
| | | TC | 16 (51.61%) | 5 (38.46%) | |
| | | CC | 13 (41.94%) | 8 (61.54%) | |
| | Additive | TT (2) | 2 (6.45%) | 0 (0.00%) | 0.1808 |
| | | TC (1) | 16 (51.61%) | 5 (38.46%) | |
| | | CC (0) | 13 (41.94%) | 8 (61.54%) | |
| | Dominant | TT or TC (1) | 18 (58.06%) | 5 (38.46%) | 0.2349 |
| | | CC (0) | 13 (41.94%) | 8 (61.54%) | |
| | Recessive | TT (1) | 2 (6.45%) | 0 (0.00%) | 1.0000 |
| | | TC or CC (0) | 29 (93.55%) | 13 (100%) | |

SNP, single nucleotide polymorphism; CR complete response; PR, partial response; SD, stable disease

**Table 6. Skin toxicity according to genotyping of IL-8.**

| SNP | Mode | | Skin toxicity (+) | Skin toxicity (-) | P-value |
|-----|------|---|-------------------|-------------------|---------|
| 251 | Codominant | AA | 3 (8.11%) | 2 (20.00%) | 0.4628 |
| | | TA | 16 (43.24%) | 3 (30.00%) | |
| | | TT | 18 (48.65%) | 5 (50.00%) | |
| | Additive | AA (2) | 3 (8.11%) | 2 (20.00%) | 0.6598 |
| | | TA (1) | 16 (43.24%) | 3 (30.00%) | |
| | | TT (0) | 18 (48.65%) | 5 (50.00%) | |
| | Dominant | AA or TA (1) | 19 (51.35%) | 5 (50.00%) | 1.0000 |
| | | TT (0) | 18 (48.65%) | 5 (50.00%) | |
| | Recessive | AA (1) | 3 (8.11%) | 2 (20.00%) | 0.2853 |
| | | TA or TT (0) | 34 (91.89%) | 8 (80.00%) | |
| 781 | Codominant | TT | 1 (2.70%) | 2 (20.00%) | 0.2010 |
| | | TC | 19 (51.35%) | 4 (40.00%) | |
| | | CC | 17 (45.95%) | 4 (40.00%) | |
| | Additive | TT (2) | 1 (2.70%) | 2 (20.00%) | 0.2869 |
| | | TC (1) | 19 (51.35%) | 4 (40.00%) | |
| | | CC (0) | 17 (45.95%) | 4 (40.00%) | |
| | Dominant | TT or TC (1) | 20 (54.05%) | 6 (60.00%) | 1.0000 |
| | | CC (0) | 17 (45.95%) | 4 (40.00%) | |
| | Recessive | TT (1) | 1 (2.70%) | 2 (20.00%) | 0.1101 |
| | | TC or CC (0) | 36 (97.30%) | 8 (80.00%) | |

SNP, single nucleotide polymorphism

## Discussion

The major finding of this study was that patients who developed skin rash after Cmab exposure had lower baseline IL-8 levels compared to those who did not develop skin rash or those who were not treated with Cmab. Even though the development of skin rash were associated with numerically improved median OS as previous reported, these findings were not statistically significant due to small number of the patients. We enrolled patients who were treated with Bmab as a negative control.

We hypothesized that IL-8 could play a critical role in the development of skin toxicity in patients treated with Cmab. In the present study, we showed that low levels of serum IL-8 prior to Cmab exposure in patients were associated with development of skin toxicity. A previous *in vitro* study demonstrated that EGFR inhibition resulted in decreased IL-8 expression in keratinocytes. In patients treated with various EGFR inhibitors, including gefitinib, erlotinib, cetuximab, and panitumumab, low levels of serum IL-8 correlating with stronger EGFR inhibition were also associated with a higher grade of skin toxicity [9].

IL-8 is a member of the CXC chemokine family, which is known to attract neutrophils and lymphocytes [25, 26]. A wide range of normal and tumor cells can express IL-8, and the important role IL-8 plays is to initiated and magnify the acute inflammatory response [27]. In addition, several reports have shown that IL-8 plays a role in the pathogenesis of cancer, including angiogenesis, growth, and metastasis [28–32].

Bangsgaard et al. reported that the neutralization of IL-8 prevented skin toxicity associated with EGFR inhibitors [12]. This study suggests that topical manipulation of IL-8 may be a potential target for Cmab-induced skin rash without affecting the systemic efficacy of the treatment. Additionally, EGFR inhibition is known to reduce IL-8 expression. Paul et al. reported

that elevated CCL2 and CCL5 levels, and decreased IL-8 expression were detected in keratino-cytes after EGFR inhibition [9]. In patients treated with erlotinib, a lower serum level of IL-8, leading to stronger EGFR inhibition, was associated with a higher grade of skin toxicity. These results are in line with the present study: In patients treated with Cmab, a low level of serum IL-8 was associated with skin toxicity.

Based on our results, IL-8 levels may serve as a predictive marker of Cmab-induced skin toxicity. Skin toxicity is one of the major adverse events in patients treated with Cmab. This toxicity adversely affects patients' quality of life and treatment compliance. Therefore, it would be useful to predict high-risk patients who are susceptible to Cmab skin toxicity. A sub-sequent study would lead us to classify patients according to the serum level of IL-8 prior to treatment.

In addition, the mean serum IL-1β level at 24 h after cycle 1 in patients with skin toxicity was lower than that in patients without skin toxicity. Previous reports have shown that the inhibition of EGFR induces IL-1 and tumor necrosis factor-alpha in mice [33]. These chemo-kines induce IL-8 secretion by fibroblasts and keratinocytes, leading to neutrophil migration in cutaneous tissue [34–36].

Several studies have been conducted to investigate the relationship between IL-8 gene poly-morphisms and the risk of developing various types of cancers including gastric, breast, lung, colon, and ovarian cancer. Previous studies have shown that IL-8 polymorphisms may affect IL-8 levels [22–24]. However, IL-8 genotypes were not associated with IL-8 levels in our study. Owing to the small sample size, we were unable to show a strong association between IL-8 polymorphisms and skin toxicity. IL-8 polymorphisms were not related to tumor response. To the best of our knowledge, our study is the first to focus on whether IL-8 polymorphisms might affect skin toxicity and tumor response in mCRC patients treated with Cmab-based chemotherapy.

## Conclusion

In conclusion, our results showed that the serum level of IL-8 was lower in mCRC patients with skin toxicity after Cmab treatment than in those without skin toxicity. Furthermore, IL-8 genotypes were not associated with skin toxicity or tumor responses. Given that skin toxicity, which is a host response to EGFR inhibition, has a prognostic value in patients treated with EGFR inhibitors, we showed a significant correlation between serum IL-8 concentrations and the severity of skin rash ($p = 0.0341$). IL-8 levels may serve as a functional biomarker for effec-tive EGFR inhibition in the future.

## Supporting information

**S1 Dataset.**
(XLSX)

**S1 Raw data.**
(ZIP)

## Acknowledgments

The results were partly presented at the 2019 Annual Meeting of the American Association of Cancer Research. We would like to thank Dr. Sohee Oh for their help at the Medical Research Collaborating Center, SNU-SMG Boramae Medical Center.

## Author Contributions

**Conceptualization:** Jin Hyun Park, Mi Young Kim, In Sil Choi, Ji-Won Kim, Jin Won Kim, Keun-Wook Lee, Jin-Soo Kim.

**Data curation:** Jin Hyun Park, Mi Young Kim, In Sil Choi, Ji-Won Kim, Jin Won Kim, Keun-Wook Lee, Jin-Soo Kim.

**Formal analysis:** Jin Hyun Park, Mi Young Kim, In Sil Choi, Ji-Won Kim, Jin Won Kim, Keun-Wook Lee, Jin-Soo Kim.

**Funding acquisition:** Jin-Soo Kim.

**Investigation:** Jin Hyun Park, Mi Young Kim, Jin Won Kim, Keun-Wook Lee, Jin-Soo Kim.

**Methodology:** Jin Hyun Park, Mi Young Kim, Jin-Soo Kim.

**Project administration:** Jin Hyun Park, Jin-Soo Kim.

**Resources:** Jin Hyun Park, Jin-Soo Kim.

**Software:** Jin Hyun Park, Jin-Soo Kim.

**Supervision:** Jin Hyun Park, Jin-Soo Kim.

**Validation:** Jin Hyun Park, Jin-Soo Kim.

**Visualization:** Jin Hyun Park, Jin-Soo Kim.

**Writing – original draft:** Jin Hyun Park.

**Writing – review & editing:** Jin Hyun Park, In Sil Choi, Ji-Won Kim, Jin Won Kim, Keun-Wook Lee, Jin-Soo Kim.

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
