## [Decision Letter · Decision Letter 0]

18 May 2022

PONE-D-21-24405

Identification of immune-related mechanisms of cetuximab induced skin toxicity in colorectal cancer patients

PLOS ONE

Dear Dr. Kim,

Thank you for submitting your manuscript to PLOS ONE. After careful consideration, we feel that it has merit but does not fully meet PLOS ONE’s publication criteria as it currently stands. Therefore, we invite you to submit a revised version of the manuscript that addresses the points raised during the review process.

We look forward to receiving your revised manuscript.

Kind regards,

Rama Krishna Kancha

Academic Editor

PLOS ONE

Journal Requirements:

Additional Editor Comments (if provided):

The reviewer is of the opinion that the manuscript need to be revised. Please find the reviewer's comments attached.

Reviewers' comments:

Reviewer's Responses to Questions

**Comments to the Author**

1. Is the manuscript technically sound, and do the data support the conclusions?

Reviewer #1: Partly

2. Has the statistical analysis been performed appropriately and rigorously? 

Reviewer #1: Yes

3. Have the authors made all data underlying the findings in their manuscript fully available?

Reviewer #1: Yes

4. Is the manuscript presented in an intelligible fashion and written in standard English?

Reviewer #1: Yes

5. Review Comments to the Author

Reviewer #1: Authors performed an interesting analysis regarding a potential mechanism at the basis of anti-EGFR related skin toxicity. The paper is well written although some english typo have to be assessed.

Considering the small sample size of patients it would be interesting describe if patients with high grade skin toxicity have experienced a better survival compared with patients with no or low grade skin toxicity that received the same treatment (folfiri cetuximab).

Moreover, besides IL-8, have some other changes in the other 19 cytokines been identyfied?

6. PLOS authors have the option to publish the peer review history of their article (what does this mean?). If published, this will include your full peer review and any attached files.

Reviewer #1: No

---

## [Author Response · Author response to Decision Letter 0]

24 Aug 2022

The response is attached in the revision submission. 

Answer: Thank you very much for your comments. We had an English editing before we submitted the manuscript last year. As you mentioned, the number of the patients in this study was small (N = 38), and only 30 patients were exposed to cetuximab based chemotherapy. When we compared the survival according to the skin rash, there was no significant difference. We believe that we do not have statistical power due to small number of the patients.

---

## [Decision Letter · Decision Letter 1]

30 Aug 2022

PONE-D-21-24405R1Identification of immune-related mechanisms of cetuximab induced skin toxicity in colorectal cancer patientsPLOS ONE

Dear Dr. Kim,

Thank you for submitting your manuscript to PLOS ONE. After careful consideration, we feel that it has merit but does not fully meet PLOS ONE’s publication criteria as it currently stands. Therefore, we invite you to submit a revised version of the manuscript that addresses the points raised during the review process.

Please include the following items when submitting your revised manuscript:A rebuttal letter that responds to each point raised by the academic editor and reviewer(s). You should upload this letter as a separate file labeled 'Response to Reviewers'.A marked-up copy of your manuscript that highlights changes made to the original version. You should upload this as a separate file labeled 'Revised Manuscript with Track Changes'.An unmarked version of your revised paper without tracked changes. You should upload this as a separate file labeled 'Manuscript'.If applicable, we recommend that you deposit your laboratory protocols in protocols.io to enhance the reproducibility of your results. Protocols.io assigns your protocol its own identifier (DOI) so that it can be cited independently in the future. For instructions see: https://journals.plos.org/plosone/s/submission-guidelines#loc-laboratory-protocols. Additionally, PLOS ONE offers an option for publishing peer-reviewed Lab Protocol articles, which describe protocols hosted on protocols.io. Read more information on sharing protocols at https://plos.org/protocols?utm_medium=editorial-email&utm_source=authorletters&utm_campaign=protocols.

We look forward to receiving your revised manuscript.

Kind regards,

Rama Krishna Kancha

Academic Editor

PLOS ONE

Journal Requirements:

Additional Editor Comments (if provided):

The reviewer opines that the comments/suggestions were not rebutted completely. Kindly make a point-wise response to the reviewer's comments.

Reviewers' comments:

Reviewer's Responses to Questions

**Comments to the Author**

1. If the authors have adequately addressed your comments raised in a previous round of review and you feel that this manuscript is now acceptable for publication, you may indicate that here to bypass the “Comments to the Author” section, enter your conflict of interest statement in the “Confidential to Editor” section, and submit your "Accept" recommendation.

Reviewer #1: (No Response)

2. Is the manuscript technically sound, and do the data support the conclusions?

Reviewer #1: (No Response)

3. Has the statistical analysis been performed appropriately and rigorously? 

Reviewer #1: (No Response)

4. Have the authors made all data underlying the findings in their manuscript fully available?

Reviewer #1: (No Response)

5. Is the manuscript presented in an intelligible fashion and written in standard English?

Reviewer #1: (No Response)

6. Review Comments to the Author

Reviewer #1: Authors have not answered to a point by point review.

Could they please give comments regarding my specific questions? even if the number of patients is low I would like to see what I have requested.

Otherwise I can not accept the paper for publication

7. PLOS authors have the option to publish the peer review history of their article (what does this mean?). If published, this will include your full peer review and any attached files.

Reviewer #1: No

---

## [Author Response · Author response to Decision Letter 1]

15 Sep 2022

As the reviewer suggested, now we added the survival analyses according to the presence or grades of skin rash in the manuscript (page 9, line number 166 – 172). We are also providing the Kaplan-Meier survival curves to respond the reviewer’s comments properly (please see the attached document)

Another comment from the reviewer was as follows:

“Moreover, besides IL-8, have some other changes in the other 19 cytokines been identified?”

Answer: In the table 3, we already showed the changes of 20 cytokines levels in detail. Besides IL-8, we highlighted the changes with statistical significance in the manuscript (page 11, line number 181 – 175). the mean serum IL-1 alpha and MIP-1 beta levels for patients in group 3 (bevacizumab treated) was lower, but there was no difference between patients with skin rash and those without skin rash.

---

## [Decision Letter · Decision Letter 2]

10 Oct 2022

Identification of immune-related mechanisms of cetuximab induced skin toxicity in colorectal cancer patients

PONE-D-21-24405R2

Dear Dr. Kim,

We’re pleased to inform you that your manuscript has been judged scientifically suitable for publication and will be formally accepted for publication once it meets all outstanding technical requirements.

Kind regards,

Rama Krishna Kancha

Academic Editor

PLOS ONE

Additional Editor Comments (optional):

Reviewers' comments:

Reviewer's Responses to Questions

**Comments to the Author**

1. If the authors have adequately addressed your comments raised in a previous round of review and you feel that this manuscript is now acceptable for publication, you may indicate that here to bypass the “Comments to the Author” section, enter your conflict of interest statement in the “Confidential to Editor” section, and submit your "Accept" recommendation.

Reviewer #1: All comments have been addressed

2. Is the manuscript technically sound, and do the data support the conclusions?

Reviewer #1: Yes

3. Has the statistical analysis been performed appropriately and rigorously? 

Reviewer #1: Yes

4. Have the authors made all data underlying the findings in their manuscript fully available?

Reviewer #1: Yes

5. Is the manuscript presented in an intelligible fashion and written in standard English?

Reviewer #1: Yes

6. Review Comments to the Author

Reviewer #1: Authors have addressed all my comments and answered to my specific question regarding an analysis of the study.

Thank you.

7. PLOS authors have the option to publish the peer review history of their article (what does this mean?). If published, this will include your full peer review and any attached files.

Reviewer #1: No

---

## [Editor Report · Acceptance letter]

14 Oct 2022

PONE-D-21-24405R2 

Identification of immune-related mechanisms of cetuximab induced skin toxicity in colorectal cancer patients 

Dear Dr. Kim:

I'm pleased to inform you that your manuscript has been deemed suitable for publication in PLOS ONE. Congratulations! Your manuscript is now with our production department. 

Kind regards, 

on behalf of

Dr. Rama Krishna Kancha 

Academic Editor

PLOS ONE